# RNA Sequencing of Whole Blood Defines the Signature of High Intensity Exercise at Altitude in Elite Speed Skaters

**DOI:** 10.3390/genes13040574

**Published:** 2022-03-24

**Authors:** Andrey S. Glotov, Irina E. Zelenkova, Elena S. Vashukova, Anna R. Shuvalova, Alexandra D. Zolotareva, Dmitrii E. Polev, Yury A. Barbitoff, Oleg S. Glotov, Andrey M. Sarana, Sergey G. Shcherbak, Mariya A. Rozina, Victoria L. Gogotova, Alexander V. Predeus

**Affiliations:** 1Laboratory of Biobanking and Genome Medicine, Institute of Translational Biomedicine, Saint Petersburg State University, Universitetskaya nab., 7-9-11, 199034 St. Petersburg, Russia; 2Division of Genome Medicine, D.O. Ott Research Institute of Obstetrics, Gynecology and Reproductology, Mendeleevskaya 3, 199034 St. Petersburg, Russia; vi_lena@list.ru (E.S.V.); shuvalova_anna_radionovna@mail.ru (A.R.S.); alexzandra-x@mail.ru (A.D.Z.); brantoza@gmail.com (D.E.P.); barbitoff@bioinf.me (Y.A.B.); olglotov@mail.ru (O.S.G.); 3GENUD Research Group, FIMS Collaborating Center of Sports Medicine, Department of Physiatry and Nursing, University of Zaragoza, 50009 Zaragoza, Spain; iz@i1.ru; 4I.M. Sechenov First Moscow State Medical University (Sechenov University), Ministry of Health of Russia, 119435 Moscow, Russia; 5Bioinformatics Institute, Kantemirovskaya St., 2A, 194100 St. Petersburg, Russia; 6Children’s Scientific and Clinical Center for Infectious Diseases of the Federal Medical and Biological Agency, 197022 St. Petersburg, Russia; 7City Hospital #40, Sestroretsk, Borisov Str., 9, 197706 St. Petersburg, Russia; asarana@mail.ru (A.M.S.); sgsherbak@mail.ru (S.G.S.); 8Department of Postgraduate Medical Education, St. Petersburg State University, Universitetskaya nab., 7-9-11, 199034 St. Petersburg, Russia; 9Russian Skating Union, Luzhnetskaya nab., 8, Office 230, 119991 Moscow, Russia; mar-rozina@yandex.ru (M.A.R.); vgogotova@gmail.com (V.L.G.)

**Keywords:** RNA-seq, RNA sequencing, exercise, expression profiling, whole blood, elite athletes, speed skating, high altitude adaptation, live high, train high (LHTH), platelets

## Abstract

Although high altitude training has been increasingly popular among endurance athletes, the molecular and cellular bases of this adaptation remain poorly understood. We aimed to define the underlying physiological changes and screen for potential biomarkers of adaptation using transcriptional profiling of whole blood. Seven elite female speed skaters were profiled on the 18th day of high-altitude adaptation. Whole blood RNA-seq before and after an intense 1 h skating bout was used to measure gene expression changes associated with exercise. In order to identify the genes specifically regulated at high altitudes, we have leveraged the data from eight previously published microarray datasets studying blood expression changes after exercise at sea level. Using cell type-specific signatures, we were able to deconvolute changes of cell type abundance from individual gene expression changes. Among these were *PHOSPHO1*, with a known role in erythropoiesis, and *MARC1* with a role in endogenic NO metabolism. We find that platelet and erythrocyte counts uniquely respond to altitude exercise, while changes in neutrophils represent a more generic marker of intense exercise. Publicly available data from both single cell atlases and exercise-related blood profiling dramatically increases the value of whole blood RNA-seq for the dynamic evaluation of physiological changes in an athlete’s body.

## 1. Introduction

Systemic effects of exercise have drawn substantial interest from researchers in medical, public health, and athletic fields for over a century [1,2,3]. In one of the earliest publications on the subject, Larrabee in 1902 described leukocytosis (a decrease in white blood cell count) in runners tested after The Boston Marathon [4]. With medical advances of the 20th century, it quickly became apparent that physical activity and sufficient exercise is the single most significant predictor of general population health and wellbeing [5]. At the same time, biological reasons for this remained elusive, and key molecular drivers of adaptation to physical exercise are yet to be defined on the systemic level. The worldwide epidemic of obesity, and the general shift towards a sedentary lifestyle, especially aggravated during the current SARS-CoV-2 pandemic, make these goals ever more important [5]. With an onslaught of the omics methods of the 21st century, we gained access to incredibly valuable genome-wide datasets, characterizing gene expression, proteome, and metabolic changes associated with exercise. However, as often happens with revolutionary methods, our understanding of the data still lags far behind.

Among the omics approaches used to characterize the effect of physical exertion on the human body, gene expression measurement has become the most popular. This happened for several reasons. Since first microarray experiments, and then with RNA sequencing (RNA-seq), gene expression methods have reached relative maturity, currently allowing the measurement of all expressed human genes in a particular tissue with high accuracy and reproducibility [6,7]. They require a modest amount of biological material that can be effectively preserved without freezing, which becomes critical in many environments. Finally, both microarray and RNA-seq experiments can be performed at a reasonable cost, making them accessible to many laboratories. Not all of these criteria are satisfied in the case of unbiased proteomic or metabolomic profiling, both of which are still in the phase of active method development [8,9].

The choice of the profiled tissue is often defined by the biological questions posed. While obesity-focused studies have often used adipose tissue biopsies [10], studies in athletes preferably profiled muscle tissue [11,12,13]. Both of these approaches, however, include relatively traumatic biopsies, making them hard to perform outside of medical facilities. Peripheral blood sampling, on the other hand, has many advantages—such as well-established collection, storage, and transportation protocols, relatively minor effects on the subject’s well-being, and the presence of many valuable biomarkers. Blood also represents a very diverse mixture of cells, giving access to an intriguing interface between metabolic and immune functions. Indeed, it is well documented that blood composition dynamically reacts to both physical and immunological challenges [14]. Because of these factors, gene expression of peripheral blood is used extremely widely, currently listing thousands of published studies. Depending on the study goals, researchers have profiled whole blood, white blood cells (WBCs), peripheral blood mononuclear cells (PBMCs), or sorted subpopulations of leukocytes [15].

More specifically, the effects of short- and long-term exercise on peripheral blood gene expression have been reported in over 50 different studies, with 20+ using whole-transcriptome profiling methods reviewed in [1]. The vast majority of the published blood whole-transcriptome datasets were generated using microarray technology. The experimental designs vary substantially; time of study changes from immediately after a bout of moderate or strenuous exercise to weeks or months of regular activity; the participants also varied in level of preparation, age, and sex. We have summarized all the relevant datasets available in the literature in Appendix A, only including the studies for which the data are openly available via databases, such as Gene Expression Omnibus (GEO) or ArrayExpress. It is worth noting that no RNA sequencing datasets were available at the time of our search, and no studies involved high altitude adaptation.

In our study, we applied whole blood RNA-seq to seven elite female skaters, in order to assess the immunological and metabolic effects of high-altitude adaptation (Appendix A). To our knowledge, this is the first such attempt. We also put our results in the context of the previously published gene expression studies, and identified candidate genes driving the adaptation, as well as general physiological changes inferred from gene expression.

## 2. Materials and Methods

### 2.1. Subjects and Study Protocol

The study was conducted during the pre-season period, and each participant underwent medical evaluations including the collection of medical history. Seven female elite speed skaters were enrolled and gave their informed written consent to participate in this study. All participants underwent a medical examination and were deemed fit for training and competitive activity; none had a history of cardiovascular, pulmonary, or metabolic diseases. Height, weight, and several physiological parameters were recorded for each athlete. The average participant age was 25.0 ± 7.0 years. All patients were of Russian ethnicity. The study was cleared by the Saint Petersburg State University Ethics Review Board for human studies (decision #40 from 7 March 2012) and was performed in accordance with the Declaration of Helsinki.

Physiological measurements and sample collections were carried out during an altitude training camp that was carried out 1850 m above sea level (Font Romeu, France, 1850 m above sea level, 21 days of stay). The study was conducted during the period after the first part of the adaptation period (after 18 days from the beginning of the altitude exposure). Pre- and post-exercise data collections were carried out in the morning between 08:00 and 11:00. Key physiological and biochemical parameters are given in Appendix A.

### 2.2. Blood Sample Collection and RNA Isolation

For each participant, a 2.5 mL whole blood sample was collected before and after the exercise using the RNAgard Blood Tubes (Biomatrica, San Diego, CA, USA) according to the manufacturer’s protocol and then stored at −20 °C until further processing. Total RNA was extracted from blood using PureLink RNA Mini Kit (Thermo Fisher Scientific, Inc., Waltham, MA, USA) and «BioMaxi™ Precipitation Buffer» (Biomatrica, San Diego, CA, USA), according to the manufacturer’s protocol. RNA concentration was measured using Quantus Fluorometer TM with QuantiFluor RNA System kit (Promega, Madison, WI, USA). RNA quality control was performed using capillary gel electrophoresis on a QIAxcel Advanced System (Qiagen, Dusseldorf, Germany). Total RNA was depleted of globin mRNA with GLOBINclear—Human Kit (Invitrogen, Waltham, MA, USA) according to the manufacturer’s protocol.

### 2.3. Library Preparation and Illumina RNA Sequencing

Fourteen samples of globin-depleted whole blood (seven skaters, before and after exercise) were sequenced using strand-specific RNA-seq with polyA selection. Libraries were prepared using TruSeq Stranded mRNA Library Prep Kit (Illumina, Inc., San Diego, CA, USA) according to the TruSeq Stranded mRNA Sample Preparation Guide # 15,031,047 E (Illumina, San Diego, CA, USA). Validation of the libraries was performed on the QIAxcel Advanced System (Qiagen, Hilden, Germany). Library quantification was performed using Quantus Fluorometer with QuantiFluor dsDNA System kit (Promega, Madison, WI, USA). Paired-end sequencing of the libraries was performed on HiSeq 4000 System with a reading length of 2 × 150 bp using HiSeq 3000/4000 PE Cluster Kit and HiSeq 3000/4000 SBS Kit (300 cycles) (Illumina, San Diego, CA, USA). The number of reads obtained per sample varied from 12.4 to 40.8 M, with a mean of 31.3 M, and a median of 32.2 M per sample. Raw reads and processed data were deposited in the Gene Expression Omnibus database under study ID GSE164890.

### 2.4. Alignment and Quantification

Assessment of raw read quality was performed using FastQC v0.11.6. Paired-end reads were aligned using STAR v2.5.3a [16] to the primary assembly of the human genome (version GRCh38.p10), with GENCODE v26 annotation [17] with pseudoautosomal (PAR) gene copies removed. STAR options “--outSAMtype BAM SortedByCoordinate --quantMode TranscriptomeSAM” were enabled, thus generating alignments to both genome and transcriptome. Overall, 94.5–98.6% reads (median 96.0%) were successfully aligned, with 2.5–11.5% overall reads (median 2.7%) aligning to rRNA. Genome BAM files were used to generate TDF files using igvtools v2.3.93 and visualized using IGV v2.4.11 [18]. Transcriptomic BAM files were used for quantification with RSEM v1.2.31 [19], with “--strandedness reverse” option enabled, according to the strand-specific library preparation type, and generating expression tables in raw counts, TPM, and FPKM on a transcript and gene level. After quantification, 67.6–82.5% (median 81.9%) of the original reads were successfully assigned to the genes present in Gencode v26 annotation. A detailed pipeline for read quality control, alignment, and quantification is available at https://github.com/apredeus/rnaquant, accessed on 17 January 2022.

### 2.5. Differential Expression and Pathway Enrichment Analysis

Differential expression and overrepresentation pathway enrichment analysis, as well as all other bioinformatic analyses from this section on, was performed in *R* v4.0.4. Per-gene expression table generated by RSEM was used for differential gene expression with *DESeq2* R package v1.18.1 [20], retaining genes with FDR < 0.1. Variance-stabilizing *rlog* transformation from the *DESeq2* package was used to normalize the expression data for diagnostic plotting. Donor effect correction was performed on the rlog-transformed matrix using the *comBat* function from the *sva* R package [21]. Pathway enrichment was performed using hallmark (H) and canonical (CP) gene set collections from MsigDB [22,23] and packages *clusterProfiler* [24] and fGSEA [25].

### 2.6. Public Single Cell RNA-Seq Dataset Processing

Publicly available single cell RNA-seq datasets (GSE149938 and 10k PBMC cells from 10X Genomics) were downloaded locally and processed using the *Seurat* package [26]. Each dataset was filtered, normalized, clustered to generate coarse-grained cellular populations, and markers defining each cell type were generated.

### 2.7. Microarray Dataset Reanalysis

Eight selected microarray studies were uniformly re-processed using the *GEOquery* [27] and *limma* [28] packages. Each dataset was visualized (Appendix A), and a list of differentially expressed genes was generated using a pairwise *limma* linear model that included both donor and exercise. Detailed scripts for all performed analysis and figure generation are freely available at https://github.com/apredeus/skater_rnaseq, accessed on 17 January 2022.

## 3. Results

### 3.1. Blood Panel and Physiological Measurements

Biochemical blood parameters (see Appendix A for the full list) and physiological measurements (fat mass and percentage, muscle mass and percentage, total fluid, phase angle, tHb-mass, total circulating blood volume (TCBV), hemoglobin, hematocrit, and percent recovery index in each microcycle) were measured throughout the adaptation period and were found to be in line with previously reported values. Due to logistic restrictions, daily physiological measurements detailing the adaptation process were not performed; instead, testing was performed on several select days. At the same time, many reports detailing biochemical and physiological adaptation to high altitude have been published previously and were not the aim of this study [29,30].

Whole blood gene expression was measured in samples collected before and after a morning bout of strenuous exercise on day 18 of adaptation. The mean running time in the exercise tests was 39.0 ± 14.8 min. Heart rate was 182 ± 3 bpm at the end of the exercise. At the end of the exercise, lactate concentrations were significantly increased (3.4 ± 0.7 vs. 1.1 ± 0.2 mmol/L; *t*-test *p* < 0.05). Biochemical parameters immediately before and after the exercise were taken for six markers: total phosphate, cortisol, growth hormone, total testosterone, total T4 hormone, and CPK. Using paired Wilcoxon test we have evaluated the significance of the observed changes. We have found that cortisol and phosphate significantly decreased after the exercise, while CPK and growth hormone have increased; total T4 and testosterone remained unchanged (Appendix A).

### 3.2. Differential Gene Expression Analysis

After read alignment and quantification (see Methods), we have performed the initial evaluation of the dataset. Using 18,000 most expressed genes and principal components analysis (PCA) plot (Figure 1a), we have assessed the difference between samples collected before and after exercise. A clear donor effect was visible from the plot, with samples belonging to the same donor being closer to each other than to samples of the same group. Thus, we have applied linear donor correction using the *comBat* function of the *sva* R package. The PCA plot after the correction (Figure 1b) has shown a much clearer separation of groups. This is common in studies of human blood in particular because blood composition varies notably between individuals. From this analysis, we conclude that it is beneficial to include donor covariate in all subsequent analyses.

Differential expression analysis has uncovered substantial changes in gene expression, with 2516 genes up- and 1542 down-regulated (see Appendix A for a full list of genes). The difference between the up- and down-regulated gene numbers becomes more pronounced when looking at highly expressed genes; for example, when only genes with a mean TPM of 100 or more were considered, 582 up-regulated and 55 down-regulated genes remained (Figure 1d). For most differentially expressed genes, expression change magnitude was modest (Figure 1c,e): only 104 up-regulated and 58 down-regulated genes changed their expression more than twofold.

### 3.3. Pathway and Functional Category Analysis

After the initial assessment of differentially expressed genes, we aimed to dissect the functional and molecular pathways regulated by the exercise. To this end, we used the molecular signature database (MsigDB) pathway collection of annotated pathways relevant to human biology, immunology, metabolism, and disease [22,23]. We used overrepresentation analysis with hallmark (H) and canonical pathways (CP) gene set collections to define the major biological categories of interest. Figure 2a–c shows the top 10 representative pathways ranked by significance. Overall, inflammatory and immune pathways dominated the observed changes, with TNFa/Nf-kB, complement, interferon, IL6/JAK/STAT3 and other pathways showing strong up-regulation. On a cellular level, neutrophil degranulation and platelet activation account for a substantial fraction of up-regulated genes. Finally, hypoxia-related genes and angiogenesis via VEGFA/EGFR2 are also strongly up-regulated in response to exercise.

Among the down-regulated genes, only one hallmark gene set (HALLMARK MYC TARGETS V2) was determined to be significant. *MYC* is a well-known blood oncogene that is particularly important in lymphomas [31] and has a crucial influence on cell survival and proliferation. *MYC* gene itself was also significantly down-regulated after the exercise (Appendix A). Together with the up-regulation of the apoptosis pathway (Figure 2a), we can hypothesize that a fraction of blood cells undergo apoptosis in response to vigorous exercise, which has been described before [32]. The majority of other pathways enriched among the down-regulated genes were related to ribosomal proteins and other components of transcriptional and translational machinery (Figure 2c). An interesting standout is the DNA repair pathway, which also appears to be down-regulated alongside its most famous member, TP53.

Since most of the observed gene expression changes were modest, gene set enrichment analysis (GSEA) may have offered additional insights, potentially highlighting metabolic processes obscured by more pronounced immune gene changes [33]. To our surprise, however, GSEA analysis results were in exceptional agreement with the simple overrepresentation analysis (Figure 2d–f). The inclusion of broader reference gene sets, such as C2 or C7, also contributed little new biological information that was not discovered using H or CP pathways (Appendix A).

### 3.4. Analysis of Cell Type Composition Changes Based on Expression Signatures

While whole blood is a very rich source of metabolic and immune markers, it represents a complex tissue that is composed of numerous cell types. It is thus hard to separate effects of changes in blood cellular composition and gene expression changes within the individual cell types, both of which influence the observed differential gene expression in the bulk sample. In order to separate the two effects, we decided to leverage publicly available single cell RNA-seq datasets, generating unique expression signatures for main cell types present in whole blood. Many published datasets use human peripheral blood mononuclear cells (PBMCs). These cells, however, represent only a minor fraction of whole blood, which contains a large number of erythrocytes, platelets, and granulocytes.

Thus, we have used a publicly available scRNA-seq dataset that profiled whole blood, GSE149938 [34], to define unique cell type markers for erythrocytes and neutrophils. Additionally, we have used the 10 k PBMC cells dataset from 10× Genomics [35] to define cell type markers of coarse-grained immune cell populations. Overall, we have defined markers of 12 cell types: erythrocytes, platelets, neutrophils, natural killer cells, plasmacytoid and myeloid dendritic cells, B cells, CD4+ memory T cells, CD4+ naive T cells, CD8+ (cytotoxic) T cells, CD14+ monocytes, and CD16+ monocytes. We have then leveraged the top 20 most discriminating markers of each cell type as a proxy allowing us to estimate the changes in a particular cell type population from the bulk RNA-seq expression profile (Figure 3). A full list of all markers is available in Appendix A.

The approach was surprisingly successful, clearly identifying putative changes in individual cell type populations. Three cell types remained relatively constant: non-classical (CD16+) monocytes, naive CD4+ T cells, and myeloid dendritic cells (Figure 3b). Populations of memory CD4+ T cells, cytotoxic CD8+ T cells, natural killer (NK), B cells, and plasmacytoid dendritic cells (pDC) were decreased, albeit the latter could be defined with the least confidence due to the extreme rarity of pDCs in peripheral blood. On the other hand, populations of neutrophils, CD14+ monocytes, erythrocytes, and platelets notably increased after exercise. This agrees with neutrophil and platelet activation pathways up-regulation in our gene set analysis (Figure 2b,e).

### 3.5. Context of Other Exercise Expression Datasets

In our experiments, we have compared whole blood transcriptome profiles before and after exercise, at the expected peak of high-altitude adaptation, 18 days since the beginning of the training camp. In order to assess the influence of high-altitude adaptation, it would have been necessary to conduct similar experiments after a similar training camp at low altitudes. Unfortunately, such a comparison could not be performed due to logistic restrictions. In order to find the genes uniquely regulated at high altitude, and to put our data in the context of the previously published expression datasets, we have compiled a collection of relevant publicly available data (Appendix A). The following criteria had to be satisfied for inclusion: (1) whole transcriptome profiling using microarray or RNA-seq; (2) dataset is publicly available; (3) whole blood, white blood cells, or PBMCs profiled; (4) samples from the same donor before and after a period of intense exercise were available. We have selected eight such datasets [36,37,38,39,40,41,42,43]. The full list of considered datasets is provided in Appendix A.

We have then re-analyzed the selected datasets using the *GEOquery* and *limma* packages (see Methods). All but one of the reprocessed datasets have shown pronounced donor effects, which had to be accounted for in the fitted linear model, and significant separation of before- and after- exercise groups was observed after the donor correction (Appendix A). The outlier dataset [42] was removed from further analysis. We have then filtered the differentially expressed genes identified in at least one of seven datasets from our analysis, in order to identify the genes specific to the high-altitude adaptation. Taking a conservative approach, we removed all genes deemed significant in previous studies using the unadjusted *p*-value. This filtering removed 65% of the differentially expressed RNA-seq genes, leaving 685 up-regulated and 795 down-regulated genes (Figure 4a).

We next tested whether the genes uniquely up- or down- regulated in our dataset have significant enrichment of marker genes for individual blood cell types explored in the previous section. We utilized Fisher’s exact test to compare the proportion of cell type markers among all DEGs and among the uniquely regulated DEGs. We discovered a dramatic enrichment of platelet marker genes in the set of altitude-specific DEGs, with up to 90% of all differentially expressed platelet markers being uniquely regulated in our dataset (*p*-value = 4 × 10^−22^) (Figure 4b). Erythrocytes were the only other cell type that showed a significant enrichment, though to a much lower extent (*p*-value = 4 × 10^−12^). Neutrophil marker genes, on the other hand, were not enriched among the unique DEGs despite a very strong enrichment of neutrophil markers among all DEGs (Figure 4b). These results suggest that platelets and (to a lesser extent) erythrocytes are the two major cell types that uniquely respond to altitude exercise, while neutrophils are a more generic marker of intense exercise.

All of the used published transcriptomic datasets were microarray experiments, which are known to have a lower dynamic range and sensitivity than RNA-seq [7]. Thus, to reliably define a gene signature of exercise in high-altitude adaptation and provide additional validation of enrichment results, we have additionally prioritized differential genes by expression (TPM ≥ 10) and magnitude of the regulation (absolute log2FC ≥ 0.5). This resulted in a list of 72 genes (53 up- and 19 down-regulated). We were satisfied to discover that gene expression of cell-specific markers perfectly segregated with directions of cell populations changes defined above (Figure 3). Concordantly with our earlier observations, platelet marker genes (including the canonical *PPBP* marker) were also significantly enriched among the highly expressed altitude-specific DEGs (*p*-value = 2 × 10^−4^). In addition to platelets, the biggest cell-type specific changes were associated with increases in neutrophils, CD14+ macrophages, and a decrease in natural killer cells (Figure 4c). Aside from these, 27 genes (18 up- and 9 down-regulated) were not directly attributable to any specific cell type.

Several genes specific to high-altitude exercise (Figure 4c) stand out upon closer examination of available literature. One of the most up-regulated genes, *MARC1*, is a mitochondrial enzyme catalyzing the reduction of N-oxygenated molecules [44] that was postulated to influence the levels of endogenous nitric oxide (NO) [45]. Given the extremely broad and important role of NO in cardiovascular physiology in general [46], and in erythrocyte adaptation to hypoxia specifically [47], it seems very possible that this gene plays a key role in high altitude adaptation. Human protein atlas (proteinatlas.org, [48], accessed on 29 April 2021) show that *MARC1* mRNA is enriched in sorted granulocytes and monocytes. Thus, since up-regulation of this gene was never detected in previously published exercise studies, we can hypothesize that increased *MARC1* expression in these cell types, together with a general increase in granulocyte and CD14+ monocyte populations, can serve as a basis for physiological adaptation at high altitude.

Another gene of great interest is *PHOSPHO1*, a phosphatase that is expressed in neutrophils and eosinophils according to the Human Protein Atlas, and in neutrophils and erythrocytes according to the markers we derived from scRNAseq. Transcription of *PHOSPHO1*, which mediates the hydrolysis of phosphocholine to choline, was recently shown to be strongly upregulated during the terminal stages of erythropoiesis [49]. Up-regulation of *PHOSPHO1* caused the increased catabolism of phosphatidylcholine and phosphocholine during the terminal erythropoiesis, and its depletion caused impaired differentiation of fetal mouse and human erythroblasts. The fact that up-regulation of this gene was never detected in previous studies makes it an excellent candidate to be a key dynamic regulator of high-altitude adaptation to hypoxia.

## 4. Discussion

Sports medicine has historically been conservative and lagged behind mainstream medicine in the translation of scientific findings, often relying on “coach wisdom” or similar practices instead. This, however, has changed dramatically during the last several decades [50]. There is currently a great interest in applying modern analytical techniques to athlete health surveillance, training guidance [51], and even prohibited substances use monitoring [52]. One of the debated questions is training at high elevation. There are currently numerous strategies, such as “live high, train high” (LHTH), “live high, train low” (LHTL), and many others; however, the molecular basis of this adaptation is far from understood.

In this study, we have evaluated the results of whole blood RNA sequencing of elite female athletes and identified a significant number (over 4000) genes that are up- or down-regulated as a result of vigorous exercise after high-altitude adaptation. Given the complex cellular composition of the whole blood, biological and physiological interpretation of such profound expression changes presents a formidable task. At the same time, the progress of modern gene expression profiling methods together with the growing culture of open data sharing has allowed us to make significant strides in the interpretation of the observed changes on a cellular level. The results demonstrate that the major changes associated with altitude exercise are related to innate immune response (inflammation), hypoxic stress response, and platelet activity. Using the marker genes of different blood cell types derived from public single-cell RNA sequencing data we dissected the alterations in blood cell composition and discovered that the proportions of neutrophils, erythrocytes, CD14+ monocytes, and platelets are increased in response to exercise at altitude.

We have also leveraged a rich collection of exercise-related blood expression profiling experiments from public sources to define genes that were uniquely up- and down-regulated in our dataset. We find several notable genes that are highly expressed in blood cells and could serve as key regulatory elements responding to exercise in hypoxic conditions. These genes include *PHOSPHO1*, *MARC1*, and a number of others, including several platelet marker genes. Our analysis suggests that the majority of such platelet markers are uniquely associated with altitude exercise and are not differentially expressed in any other conditions according to published studies.

Our results provide a potential molecular link between hypoxia, platelet activity, and thrombosis. It has been long known that prolonged stay at high elevations is associated with an increased risk of thrombosis [53]. Perhaps the most surprising parallel here could be related to COVID-19, which is also associated with both hypoxia and increased risk of thrombosis. It has recently been shown that physical activity influences the outcome of COVID-19 [54]. Given that thromboses are one of the dominant causes of death in COVID-19 patients, it can be hypothesized that physical activity, especially at high altitude, may serve as the pre-conditioning factor that might alleviate the relative effects of COVID-19 and prevent the negative systemic impact of platelet hyperactivity.

## Figures and Tables

**Figure 1 genes-13-00574-f001:**
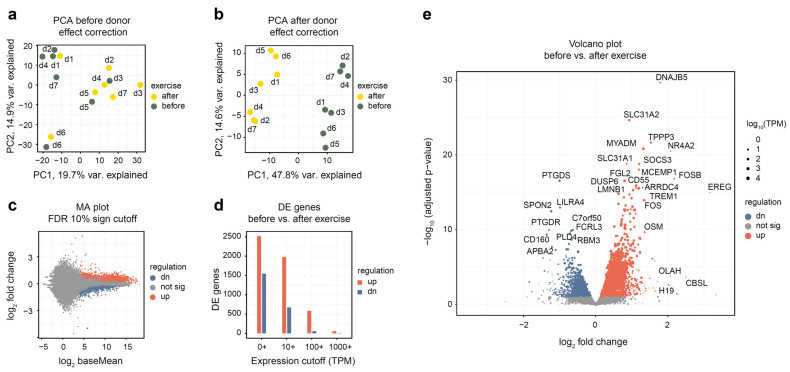
Overall assessment and differential expression analysis of whole blood RNA-seq from seven altitude-adapted female skaters before and after exercise. Differential expression analysis was performed using DESeq2 and “donor + exercise” design. Differentially expressed genes were reported at 10% FDR. “Up-regulated” indicates genes whose expression increased after the exercise. (**a**,**b**) PCA plot of the 14 studied samples, before and after donor-effect correction using *comBat*. Top 18,000 genes were used. Read counts were normalized using the rlog function from the DESeq2 package. (**c**) Log ratio—mean expression (MA) plot, with marked differentially expressed genes. (**d**) Number of differentially expressed genes depending on mean expression cutoff; TPM, transcripts per million. (**e**) Volcano plot of differentially expressed genes. Point size is scaled proportionally to mean gene expression.

**Figure 2 genes-13-00574-f002:**
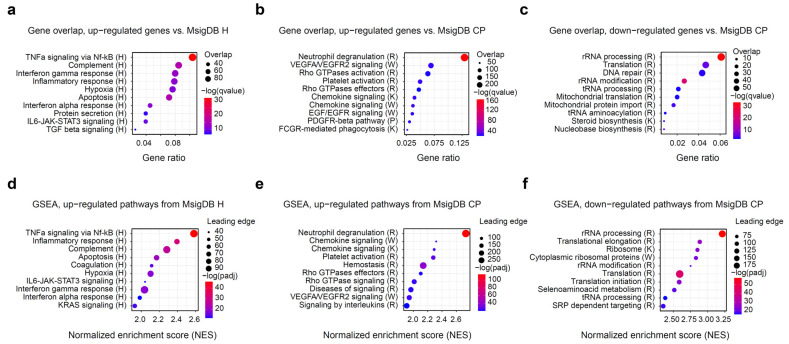
Gene overrepresentation and gene set enrichment (GSEA) analysis of the differentially expressed genes. Molecular signature database (MsigDB) H and CP collections were used to functionally characterize expression changes. Letter following the pathway name denotes its source: R, Reactome; K, KEGG; W, WikiPathways. (**a**–**c**) Top 10 significantly up- and down-regulated pathways, according to Fisher’s exact test, calculated with clusterProfiler. Down-regulated hallmark pathways only included one significant gene set (HALLMARK MYC TARGETS V2), and were omitted from the plot. (**d**–**f**) Top 10 significantly up- and down-regulated pathways according to GSEA, calculated with fGSEA. Down-regulated hallmark pathways only included one significant gene set (HALLMARK MYC TARGETS V2) and were omitted from the plot. Gene overlap indicates the number of genes in the leading edge.

**Figure 3 genes-13-00574-f003:**
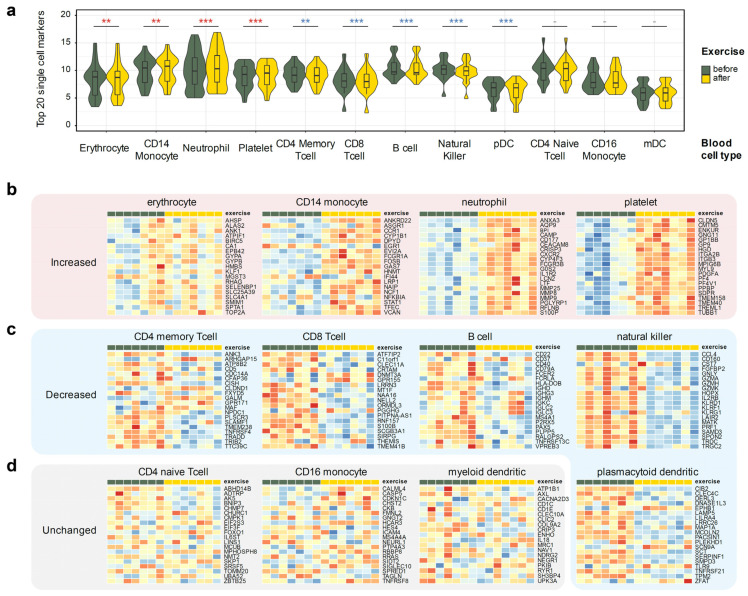
Using single cell markers to infer changes of the cell types from the whole blood RNA-seq data. Twenty markers specific to the 12 listed blood cell types were derived using two public single cell RNA-seq datasets (Methods). Donor-corrected, rlog-transformed expression matrix was used for all heatmap plots. Wilcoxon ranked sum *p*-value notation: ***, *p* < 0.001; **, *p* < 0.01; -, *p* ≥ 0.05. (**a**) Observed gene expression of 20 cell type markers in all seven profiled donors. (**b**) Cell types that increase after the exercise: neutrophils, CD14+ monocytes, platelets, and erythrocytes. (**c**) Cell types that decrease after the exercise: CD4+ memory T cells, CD8+ T cells, natural killer cells, B cells, and plasmacytoid dendritic cells. (**d**) Cell types that did not display concerted change in markers: CD4+ naive T cells, CD16+ monocytes, and myeloid dendritic cells.

**Figure 4 genes-13-00574-f004:**
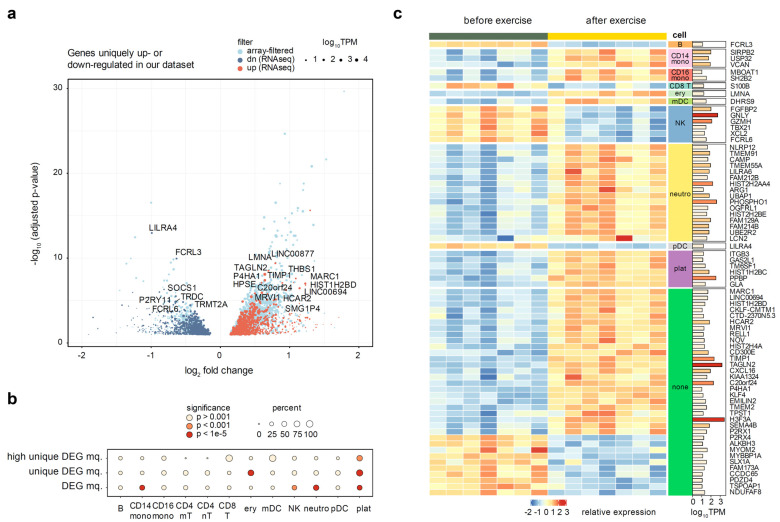
Analysis of genes uniquely up- or down-regulated in our dataset, as compared to seven public microarray datasets. Full list of datasets is given in Appendix A. (**a**) Volcano plot of genes uniquely regulated in the whole blood of altitude-adapted skaters. Light blue points indicate differentially expressed genes previously seen in at least one other exercise dataset. Point size is scaled proportionally to mean gene expression. (**b**) A circle plot representing the enrichment of cell type markers among all differentially expressed genes and altitude-specific differentially expressed genes. The size of the circle is proportional to the percentage of marker genes in the target set, and the fill of the circle corresponds to the significance levels of Fisher’s exact test. (**c**) Heatmap of highly expressed (TPM > 10) and regulated (absolute log fold change > 0.5) genes unique to our dataset. Breakdown by cell type is conducted based on single cell markers defined earlier. Rows are sorted by cell type and then by log fold change.

## Data Availability

Detailed scripts for all performed analysis and figure generation are freely available at https://github.com/apredeus/skater_rnaseq, accessed on 17 January 2022. All the data files needed to reproduce every analysis conducted in the paper are available in the repository. Raw reads and processed data were deposited in Gene Expression Omnibus database under study ID GSE164890.

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
