# Peer review of "RNA Sequencing of Whole Blood Defines the Signature of High Intensity Exercise at Altitude in Elite Speed Skaters"

_genes, 2022, doi:10.3390/genes13040574_

Round 1

Reviewer 1 Report

This is an interesting study that has good experimental design. The authors have justified the approaches used giving reasons for them. They have increased the resolution of their study by narrowing down to the cell types where gene expression changes occur. They analyzed a list of previously established marker genes for the cell types. They have also compared with data sets showing gene expression changes after exercise at low altitude.

Some questions that arise from the paper=

  1. How did they select the 18 day time point for high altitude adaptation? Some information showing the adaptation at this time point would be useful.
  2. They have used seven female athletes for this purpose. Were the athletes used in the comparison study at low altitude female subjects, or were they a mix of the two genders?
  3. The fonts on the figures are very small. Should be larger.

Reviewer 2 Report

Introduction
"intriguing interface between metabolic and 78 immune functions"
Yes, but it is also a limit. Authors must acknowledge that.

Methods
It would be nice to have a figure with the experimental design clearly explained.

Supplementary table S2 seems to be presenting cherry picked values... Why different parameters are reported and not all of them (which you surely have) since is supplementary material and there is room for them.
I also suggest to reformat the table in a more comprehensible way and highlight off-range values.

GenCode v26???
It's from 2016... this is not good for a 2022 paper.
https://www.gencodegenes.org/human/release_26.html
Lines 2604491 gencode.v26.annotation.gtf
Lines 3241007 gencode.v39.annotation.gtf
It is substantially different.
STAR too is old, 2017. This is a minor thing, but highly correlated with the annotation issue... I see also that in Table S1 GEO datasets are dated 2017 maximum... Is it possible that you missed something?

ComBat
I did not understand the method here. Why the Authors to treat subjects with a package that is intended to correct for BATCH effect?
From the sva package:
"The sva package can be used to remove artifacts in three ways: (1) identifying and estimating surrogate variables for unknown sources of variation in high-throughput experiments (Leek and Storey 2007 PLoS Genetics,2008 PNAS), (2) directly removing known batch effects using ComBat (Johnson et al. 2007 Biostatistics) and (3) removing batch effects with known control probes (Leek 2014 biorXiv)."
Batch effects are: lanes, library operators, wrong labeling issues and known problems that are NOT linked with the biological differences.
Also they stated "The PCA plot after the correction (Figure 1B) has shown much clearer separation of groups. This is common in studies of human blood in particular, because blood  composition varies notably between individuals. From this analysis, we conclude that it is beneficial to include donor covariate in all subsequent analyses." 
Where are the references? In the S1 Figure?

I acknowledge that there are methods to account for individual variation: is this the case? Maybe the authors should explain better this or provide good statistical insights.
It can be a lack of knowledge from my side, I tried to understand and justify but I couldn't. Is this the same mathematical ground of treating individuals as random effects?

My review ends here since I can't proceed since, honestly, I do not understand.

Round 2

Reviewer 2 Report

I would like to thank the Authors for the thorough clarification on the "batch effect".

Concerning the my first point, I am sorry! While reading and drafting I was thinking about the complexity of blood in terms of tissue composition and the relative changes in cell populations in various conditions, but You addressed this in the introduction and the discussion section. I forgot to delete the sentence.

"cherry pick": I understand that You may feel uncomfortable about this, but I may be not the only one reading the data in this way so You could add - in the supplementary file maybe? - the answer you gave me. 

Annotation/Software: I see. Thank you. The thing is that many clues pointed to an "old" bouncing draft (from the annotation and the dataset to the ethic board decision date): we have so little time to acknowledge all possible aspects and this publication fever - this is not the case - often lead to low quality papers.

Discussion

The discussion is well written and complete. The only thing I do not understand - or, to be honest, agree with - is the speculation towards COVID. I am more then aware about the "zeitgeist" but I find it unnecessary, nearly off-topic. I am not asking to remove, just pointing out.